# Fabrication of Transparent and Flexible Digital Microfluidics Devices

**DOI:** 10.3390/mi13040498

**Published:** 2022-03-23

**Authors:** Jianchen Cai, Jiaxi Jiang, Jinyun Jiang, Yin Tao, Xiang Gao, Meiya Ding, Yiqiang Fan

**Affiliations:** 1College of Mechanical Engineering, Quzhou University, Quzhou 324000, China; cai198666@126.com (J.C.); jiangjinyun2@163.com (J.J.); yintaoyy120@gmail.com (Y.T.); a17535873277@163.com (X.G.); ding1600615038@163.com (M.D.); 2College of Mechanical and Electrical Engineering, Beijing University of Chemical Technology, Beijing 100029, China; xsjiangjiaxi@163.com

**Keywords:** digital microfluidics, lab-on-a-chip, ITO, PET

## Abstract

This study proposed a fabrication method for thin, film-based, transparent, and flexible digital microfluidic devices. A series of characterizations were also conducted with the fabricated digital microfluidic devices. For the device fabrication, the electrodes were patterned by laser ablation of 220 nm-thick indium tin oxide (ITO) layer on a 175 μm-thick polyethylene terephthalate (PET) substrate. The electrodes were insulated with a layer of 12 μm-thick polyethylene (PE) film as the dielectric layer, and finally, a surface treatment was conducted on PE film in order to enhance the hydrophobicity. The whole digital microfluidic device has a total thickness of less than 200 μm and is nearly transparent in the visible range. The droplet manipulation with the proposed digital microfluidic device was also achieved. In addition, a series of characterization studies were conducted as follows: the contact angles under different driving voltages, the leakage current density across the patterned electrodes, and the minimum driving voltage with different control algorithms and droplet volume were measured and discussed. The UV–VIS spectrum of the proposed digital microfluidic devices was also provided in order to verify the transparency of the fabricated device. Compared with conventional methods for the fabrication of digital microfluidic devices, which usually have opaque metal/carbon electrodes, the proposed transparent and flexible digital microfluidics could have significant advantages for the observation of the droplets on the digital microfluidic device, especially for colorimetric analysis using the digital microfluidic approach.

## 1. Introduction

Compared with the conventional microfluidic devices that continuously handle the fluid flow inside of the microchannel on polymer- or silicon/glass-based microfluidic chips, the digital microfluidic chips (DMF) can precisely manipulate the discrete fluid flow (i.e., droplet) without the requirement of physical pumps, valves, or complex microchannel structures. Each droplet in the digital microfluidic approach is an isolated reaction chamber containing reagents. The precise handling of the droplet down to picoliter can be achieved with digital microfluidic devices [1]. Digital microfluidics has been widely used in various applications, such as cell manipulation [2], virus detection [3], nucleic acid amplification [4], and ion detection [5]. Several commercial digital microfluidic products were also put on the optical and biomedical market in the past few years [6].

Various droplet driving methods have been used in digital microfluidics, including magnetic [7,8], acoustic [9], gravitational [10], and electrowetting-on-dielectric (EWOD) [11]. Currently, the EWOD is the most widely used approach for digital microfluidics with the advantage of precise control of the droplet movement. The electrowetting induces the interfacial tension gradient inside of the droplet in order to trigger the droplet movement. Multiple functions can be achieved with electrowetting control, such as droplet dispensing, translocation, merging, and splitting.

The digital microfluidic devices with single-plate configuration usually consist of the substrate, electrodes, insulation dielectric layer, and the hydrophobic layer on the top surface of the devices. The substrate of the digital microfluidic device is of decisive significance for digital microfluidic devices. The substrate is usually made of glass or silicon material, with the electrodes (e.g., chromium, chromium, gold) fabricated with photolithography method [12,13], the dielectric layer is commonly fabricated with chemical vapor deposition of parlance or silicon nitride [14]. The glass/silicon-based digital microfluidics have high electrode patterns precision but are costly on materials and processing instruments. On the other hand, various low-cost approaches for digital microfluidics have also been invented, as follows: electrodes were fabricated on PCB (printed circuit board) [15]; screen-printing was used for the fabrication of carbon electrodes on thin polymer films or paper as substrate [16]; xurography or even hand-painting were also used for the low-cost digital microfluidic approach [17].

In this study, a thin, transparent, and flexible fabrication approach is proposed for digital microfluidic devices. The commercially available ITO (indium tin oxide) coated PET (polyethylene terephthalate) film was used as the substrate. Additionally, the Nd: YLF laser was used for the patterning of the conductive ITO layer on PET substrate in order to form electrodes. Finally, a thin PE (polyethylene) film was used as the dielectric layer with hydrophobic surface treatment by Rain-X water repellent.

Compared with the previous studies on digital microfluidics using the photolithography method for patterning electrodes on silicon/glass substrate, the fabricated digital microfluidic device has a total thickness of less than 200 μm and is flexible and transparent in the visible range. On the aspect of fabrication and cost, the proposed method is much more rapid, with less requirement on the highly sophisticated instrument, also with a lower cost of materials. The proposed fabrication approach for digital microfluidic devices provides a new alternative to the conventional fabrication method for digital microfluidic devices, with the advantage of transparency in the visible range, and may have significant importance for observing the colorimetric change in the droplets in some applications.

## 2. Materials and Methods

### 2.1. Materials and Instruments

The ITO-coated PET film was obtained from MSE supplies, Arizona, AZ, USA, the PET film had a thickness of 175 μm and was covered with an ITO layer of around 220 nm in thickness. The measured resistance of the ITO layer ranged from 5.6 to 6.1 Ohm/sq. The polyethylene film (Glad cling wrap, W300N, Clorox China Limited, Guangzhou, China) was used as a dielectric (insulation) layer, with a thickness of around 12 μm. Rain-X water repellent, which was used for surface treatment, was obtained from ITW GlOBAL Brands, San Luis Obispo, CA, USA. Sodium chloride was sourced from Shanghai Aladdin Biochemical Technology Co., Shanghai, China. All materials and chemicals were used as received.

Several fabrication and testing instruments were used in this study. Nd: YLF laser (LSF20D, HGTECH, Wuhan, China), with a wavelength of 1064 nm, was used for the patterning of the ITO layer on PET substrate. The laser worked at the multi-plus overlapped mode and the pulse duration was 10 ps with the repetition rate adjustable from 0 to 200 kHz. The oscilloscope used in this study was Tektronix TDS1012B (Tektronix, Inc., Beaverton, OR, USA), and the function generator (UNI-T UTG9005C) was sourced from Uni-Trend Technology Co., Ltd., Dongguan, China. ATA-2161 high-voltage amplifier is capable of amplifying the AC/DC signal with gain adjustable from 0 to 240, up to 1600 Vpp. To control the voltage supply to each of the electrodes in the proposed digital microfluidic device, a control circuit with STC 8051 microcontroller (STCmicro Technology Co., Ltd., Beijing, China) was also designed and assembled. Olympus ols5000 laser scanning confocal microscope (Olympus Corporation, Tokyo, Japan) was used to observe and measure the laser fabricated electrodes. The optical images of the system setup and fabricated devices were taken with a Nikon D3000 Digital SLR Camera (Nikon Corporation, Tokyo, Japan).

### 2.2. Fabrication

The fabrication process of the proposed digital microfluidic device is shown in Figure 1a, the PET substrate (175 μm in thickness) covered with ITO layer (~220 nm in thickness) was selectively laser-ablated for the fabrication of electrodes array. It is worth pointing out that, other than the laser ablation that is usually strong enough to directly evaporate the materials, the role of the Nd: YLF laser in this study was to induce the thermal stress between the ITO layer and PET substrate, which finally caused the ITO film fracture and ejection from the PET substrate. The following section will discuss more details of the laser ablation process.

After laser ablation to fabricate electrodes on PET substrate, a layer of 12 μm-thick PE film (i.e., cling wrap) was used to insulate the electrodes. The PE film was cut in the same size as the PET substrate and gently attached by hand. The “bonding” between the PE film and PET substrate was achieved with the help of electrostatic charge as follows: by unrolling the PE film, the PE film becomes charged by losing or gaining electrons, when in touch with another insulator (i.e., PET substrate), the electrostatic charge induces an opposite charge and bonds the two layers.

Two types of devices were designed and fabricated for the demonstration in this study, as shown in Figure 1b,c. Figure 1b shows the design and fabricated digital microfluidic device for single droplet manipulation and two droplets merging. Figure 1c shows another electrode configuration that enables droplets to move straight and turn. The electrodes in Figure 1b,c have the same dimension of 2 mm by 2 mm with 0.5 mm spacing. The Appendix A also provide video footage for the droplet movement in Figure 1b,c.

The ITO-based electrodes on PET substrate were defined by laser ablation. The fabrication process of the electrodes is shown in Figure 2a. Compared with other laser ablation methods (e.g., CO_2_ laser) that directly meltdown and vaporize the material on the radiated spot, the Nd: YLF laser only induces the tensile stress at the laser radiated spot. When the tensile stress strength is exceeded at the laser radiated spot, the ITO film will fracture and eject from the PET substrate at the laser radiated spot [18], and finally forms the desired ITO patterns on the PET substrate.

The optical image of the patterned ITO electrodes on PET substrate is shown in Figure 2b, the relatively dark area was laser scanned and the covered ITO material has been removed, the bright area is the unaffected part that the ITO layer, and PET substrate was still firmly attached. In order to effectively remove the ITO layer without damaging the PET substrate, an overlap laser scan method was used in this study. During the laser scanning process, an overlap rate of 90% was used between the adjacent pulses (two adjacent laser radiated spots had an overlap of 90%).

During laser ablation, the PET substrate unavoidably received the thermal energy, which may cause the meltdown and solidification of the PET material along the laser-scanned route, the visible trace can be found along the laser-scanned route, as shown in Figure 2b. To minimize the thermal damage to the PET substrate during laser ablation, the pulse energy was set at 1 J/cm^2^ in this study, after several trials. The repeatability of the proposed patterning method was relatively reliable, the deviation of 8 repeated experiments was less than 10%.

The system setup for the transparent and flexible digital microfluidic device is shown in Figure 3. As shown in Figure 3, a function generator was used to generate a sinusoidal signal with a frequency of 1 kHz, then the generated signal was further amplified with a high-voltage amplifier. The final output voltage was adjustable from 0 to 500 V_rms_ in this study.

To independently control the voltage supply to each electrode, a control circuit with an STC 80C51 microcontroller was used to control a series of relays (blue blocks in Figure 3) that can independently control the voltage supply to each of the ITO electrodes. The control circuit is also able to adjust the gain of the high-voltage amplifier that finally controls the voltage output. An oscilloscope was used to monitor the signal output from the function generator.

## 3. Result and Discussion

### 3.1. Surface Properties

The fabricated digital microfluidic device is shown in Figure 4, the device has a total thickness of less than 200 μm. The device is flexible (bendable) and transparent in the visual range. The insert shows the enlarged image of the laser-ablated ITO electrodes on the PET substrate.

For easier manipulation of the droplets, it is necessary to enhance the hydrophobicity on the surface of the PE film, as illustrated in Figure 1a, a layer of commercial water repellent Rain-X was sprayed on the surface of the PE film and was allowed to air-dry at room temperature in order to increase the water contact angle from 78.3° to 103.2° (as shown in Figure 5). The device fabrication is completed after surface treatment.

A laser scanning confocal microscope was used to explore the profile of the laser fabricated ITO electrodes on the PET substrate. As shown in Figure 6, the ITO electrodes are about 500 nm higher than the PET substrate (the original ITO layer had a thickness of 220 nm), which indicates that some of the PET material may have melted and been vaporized during the laser scan process. Another interesting finding is the formation of the bugles at the edge of the ITO patterns, during the laser ablation process, the material on the laser radiated spot meltdown and vaporize, and some of the PET material was ejected and re-solidified on the edge of the scan route. Such a bulge forming phenomenon after laser ablation is also commonly reported in the CO_2_ laser ablation on thermoplastics [19]. In Figure 6, the inset on the left is the image captured by an optical microscope at the gap between the electrodes, the insert on the right illustrates the formation principle of the bulges.

The UV–VIS spectrums of the ITO-covered PET substrate and the PET substrate after laser ablation (covered ITO layer has been removed) are shown in Figure 7 (measured with UV-2600i UV–VIS Spectrophotometer, Shimadzu, Kyoto, Japan). As shown in Figure 7, the absorption of the light in the visible range (380–750 nm) is relatively low, indicating the relatively good transparency of the proposed digital microfluidic device.

### 3.2. Leakage Current and Droplet Control

The leakage current density across the proposed digital microfluidic device was measured with the AC voltage supply, ranging from 250 V_rms_ to 450 V_rms_. As shown in the inset of Figure 8, two electrodes were placed on the Rain-X-covered PE film and the bottom of the PET substrate and the current was measured five times with the voltage increased from 250 V_rms_ to 450 V_rms_ with an increment of 25 V. The error bar in Figure 8 is the statistical representation of the measured leakage current density variability after five measurements at each data point. The leakage current density increased with a higher voltage supply. Generally, the leakage current is ignorable within the operation range of the proposed digital microfluidic device. The low leakage current also indicates the thin PE film is acceptable as the dielectric layer.

The measured contact angles under different AC voltage (with a frequency of 1 kHz, droplet volume of 10 μL) with three different solutions are shown in Figure 9a. The DI water, 0.1 M, and 1 M sodium chloride were used for the contact angle measurement. The 0 V shows the contact angles without the voltage supply, when the voltage was supplied and increased, the contact angles of all three of the solutions decreased almost linearly and gradually reached contact angle saturation.

The minimum droplet driving voltages with different volumes of DI water droplets under different electrode on/off time is shown in Figure 9b. The minimum driving voltage depends on many factors, including the volume of the droplet, the thickness of the dielectric layer, and the gap between the electrodes etc. In this study, we explored the minimum driving voltages that were influenced by the electrode on/off time. As previously described, the voltage supply to each of the electrodes was independently controlled by a series of relays. We found in our experiment that the electrode on/off time also significantly influenced the minimum driving voltage, as shown in Figure 9b, and a slower electrode on/off action (droplet moving slowly) required a lower driving voltage while the faster electrode on/off action (droplet moving fast) required higher driving voltage.

## 4. Conclusions

This study proposed a fabrication method for a thin, transparent, and flexible digital microfluidic device. The ITO layer on PET substrate was patterned with laser ablation for the fabrication of electrodes, a thin layer of PE film was used as the dielectric layer with the enhanced surface hydrophobicity with a commercial water repellent. The whole device has a thickness lower than 200 μm, which is easily bendable and also transparent in the visible range. In order to demonstrate the proposed fabrication technique, several digital microfluidic devices were fabricated with various droplet manipulation functions achieved. The leaking current density, the contact angle under different voltage supply, and the minimum driving voltage were also discussed in this study.

The limitation of the proposed study comes from two aspects. Compared with the conventional digital microfluidic devices that are fabricated on quartz or silicon wafer, the minimum achievable microstructures on ITO film are limited, thus, the gap between the electrodes is relatively wide and requires a higher driving voltage. In addition, compared with the conventional digital microfluidic devices, the chemical resistance of the ITO film also has limitations on some organic solvents.

The proposed transparent digital microfluidics suggests an alternative to conventional digital microfluidic devices on glass or silicon wafers. The transparency of the proposed digital microfluidic device could be of significant importance for observing the droplets inside of the device.

## Figures and Tables

**Figure 1 micromachines-13-00498-f001:**
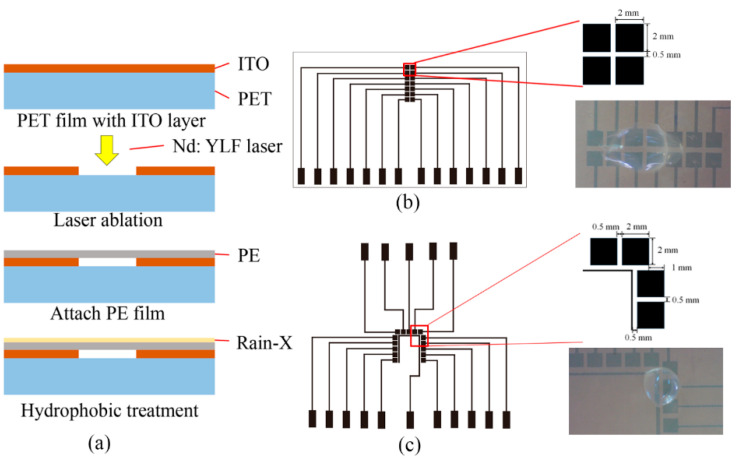
The design and fabrication process of the proposed digital microfluidic devices. (**a**): The fabrication process of the digital microfluidic device, a pulsed Nd: YAG laser was used to pattern the ITO layer on PET substrate to form electrodes array, then the ITO layer was insulated with a layer of PE film, finally, Rain-X water repellent was applied on the surface of PE film to enhance the hydrophobicity. (**b**): The digital microfluidic device for droplets merging. (**c**): The digital microfluidic device for droplets moving straight and turns. Video footages for the two proposed digital microfluidic devices are provided in the Appendix A.

**Figure 2 micromachines-13-00498-f002:**
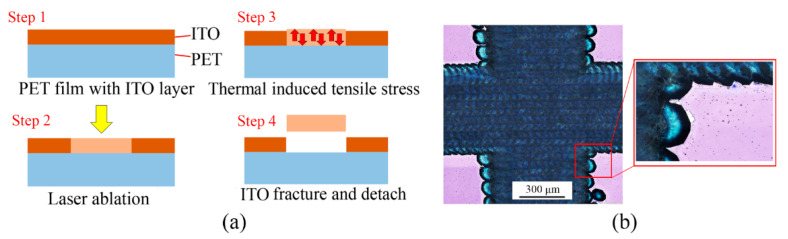
Laser ablation process on the ITO-covered PET substrate. (**a**): Laser fabrication process for patterning the ITO electrodes on PET substrate. (**b**): Optical microscope image of the laser-ablated areas on the PET substrate.

**Figure 3 micromachines-13-00498-f003:**
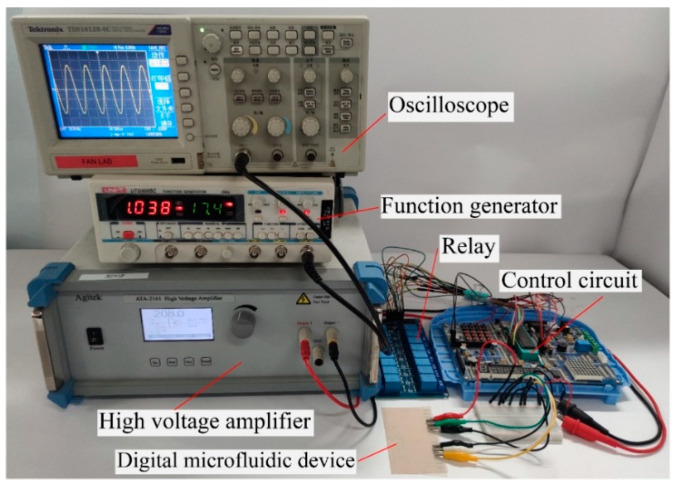
Testing system setup for the digital microfluidic device. The function generator provides the AC signal and is amplified with a high-voltage amplifier. A control circuit with microcontroller was used to control the AC voltage supply to each electrode on the digital microfluidic device.

**Figure 4 micromachines-13-00498-f004:**
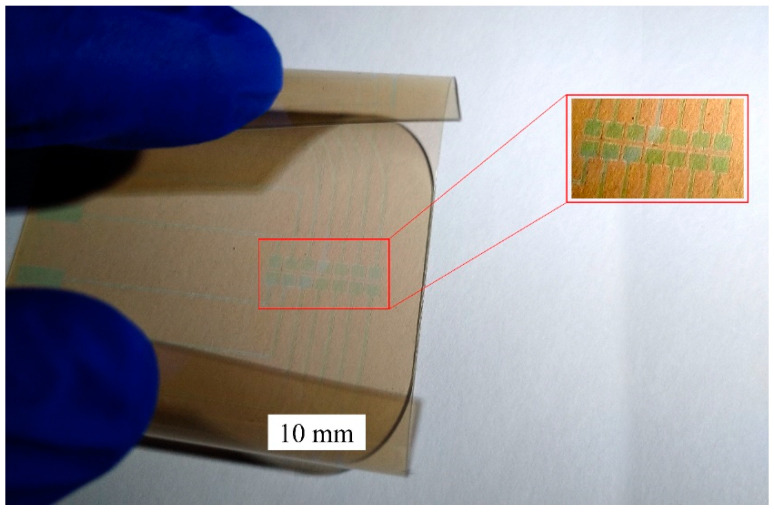
The fabricated transparent and flexible digital microfluidic device.

**Figure 5 micromachines-13-00498-f005:**
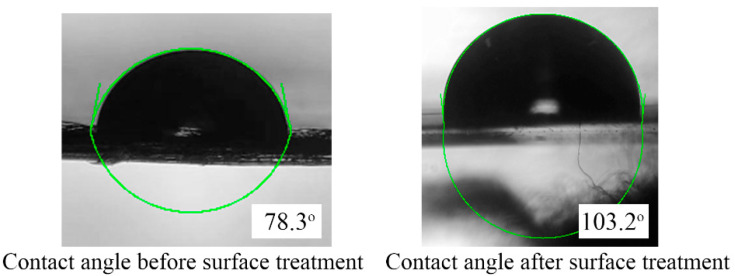
Contact angle measurement on the PE film before and after Rain-X surface treatment.

**Figure 6 micromachines-13-00498-f006:**
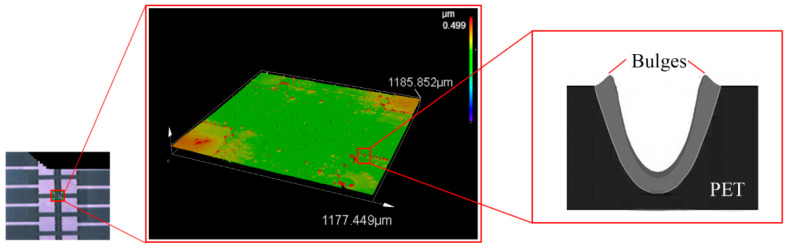
Laser confocal microscope measurement of the laser ablated surface.

**Figure 7 micromachines-13-00498-f007:**
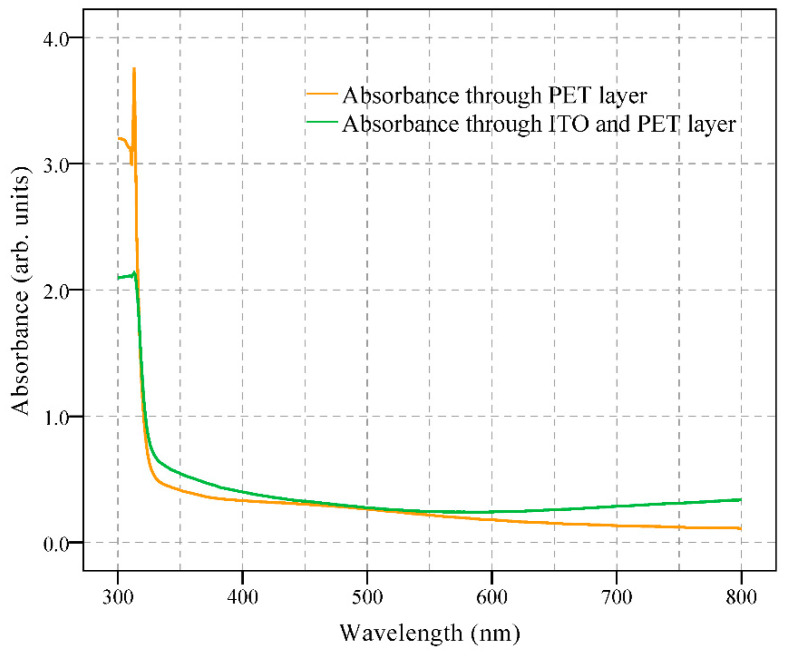
UV–VIS measurement of the PET substrate after laser ablation and ITO-covered PET substrate.

**Figure 8 micromachines-13-00498-f008:**
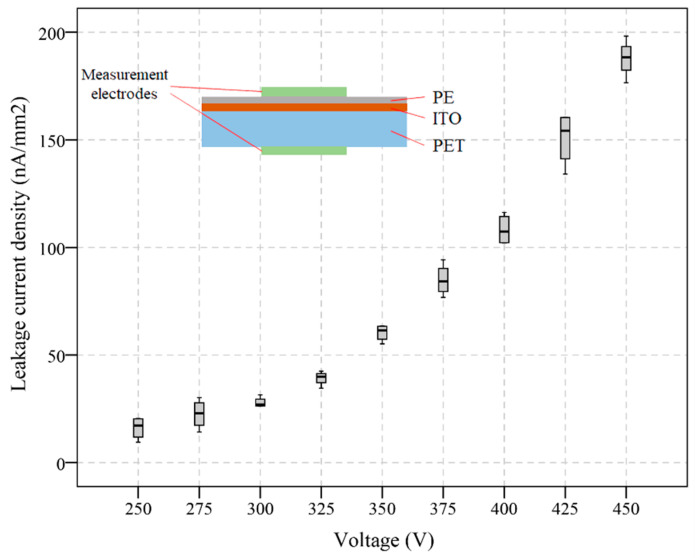
Leakage current density across the fabricated digital microfluidic device.

**Figure 9 micromachines-13-00498-f009:**
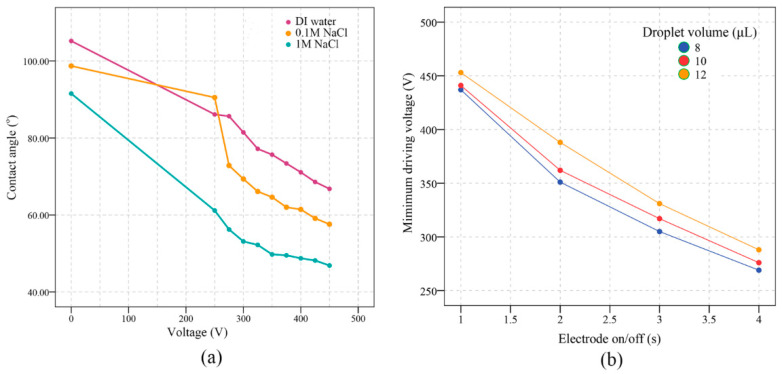
(**a**): Measured contact angle under different voltages (the frequency of the AC voltage supply is 1 kHz, with droplet volume of 10 μL). (**b**): Minimum driving voltages with the different electrode on/off time in the proposed digital microfluidic device.

## Data Availability

The data that support the findings of this study are available upon reasonable request from the authors.

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
