# Peer review of "Fabrication of Transparent and Flexible Digital Microfluidics Devices"

_micromachines, 2022, doi:10.3390/mi13040498_

Round 1
Reviewer 1 Report
This manuscript reports a fabrication process for thin-based transparent and flexible digital microfluidic device. The microfluidic device has a thickness less than 200 μm. This fabrication process could allow the development of digital microfluidic devices for colorimetric analysis. This manuscript can be improved considering the following comments:
1.-Various sentences used in the manuscript are very large. English grammar and style of all the sections of manuscript should be revised.
2.-Introduction should include more recent references from 2021 and 2022.
3.-Introduction should incorporate more information of the advantages and limitations of the main techniques used to fabricate digital microfluidic devices. In addition, authors should add the main advantages of their fabrication method to develop digital microfluidic devices in comparison with other reported in the literature.
4.-Page 2, the following sentences should have references of the different companies:
The ITO-coated PET film was obtained from MSE supplies LLC, USA.
Sodium chloride is sourced from Shanghai Aladdin Biochemical Technology Co., Ltd.
5.-Resolution of text used in figures 1-6 should be enhanced.
6.-Page 4, line 142-145, the following sentence is very large:
The system setup for the transparent and flexible digital microfluidic device is shown in figure 4, function generator was used to generate a sinusoidal signal with a frequency of 1kHz, the generated signal was further amplified with a high voltage amplifier, the final output voltage is adjustable from 0 to 500 Vrms in this study.
7.-Page 5, line 158-163, the following sentence is very large:
The process is shown in figure 5a, unlike other laser ablation method (e.g. CO2 laser) that directly meltdown and vaporize the material on the radiated spot, the Nd: YLF laser only induce the tensile stress at the laser radiated spot, when the tensile stress strength is exceeded, the ITO film will fracture and ejection from the PET substrate at the laser radiated spot [20], thus forms the desired ITO patterns on PET substrate.
8.-Page 5, line 164-170, the paragraph has two sentences very large.
9.-Resolution of figures 7-9 should be enhanced.
10.-Authors could add more discussion of the reliability of the proposed microfluidic device.
11.-Which are the main limitations or challenges of the proposed microfluidic device?
12.-What is the future research work?
13.-References should be written using the format of Micromachines.
Reviewer 2 Report
This manuscript presents a method for fabrication of transparent and flexible digital microfluidic devices. The idea seems simple and effective. However, there is no solid evidence to The introduction part lacks in-depth discussion on current research and the advantages of the proposed methods. The experimental and results were written unorganized. I would suggest the authors to re-organize the manuscript and address the following concerns.
- In the introduction part, transparent devices seem to have some advantages as the authors claimed. However, what is the benefit of being flexible?
- There is no discussion about PDMS based DMFs and soft lithography. How is PDMS-based methods and device performance compared to the proposed method/devices? PDMS is transparent, flexible and cheap. And soft lithography is also simple and can be done without using cleanroom.
- line 72, I would suggest using “Materials and Methods” instead of “Fabrication”.
- please check Line 73 and 95, repeating title
- Please use subtitle in results and discussion part for readers to better understand.
- Figure 4 and 5 should be put in experimental section, while Figure 2 and 3 are more suitable in the results section. Figure 6 is also like describing methods. I would suggest reorganizing the figures with better logic. Do not mix methods with results.
- Figure 6, please explain each graph in the caption.
- Figure 8, please explain the insert and what the datapoint and the error bars represents
- I would suggest adding result of droplet manipulation to better support the conclusion.
Round 2
Reviewer 1 Report
This manuscript was enhanced considering all the comments of reviewer.
Reviewer 2 Report
The authors made several improvements and addressed my concerns. I would recommend acceptance of this manuscript.